

# Improved salp swarm algorithm based optimization of mobile task offloading

Aishwarya R. and Mathivanan G.

Department of Computer Science and Engineering, Sathyabama Institute of Science and Technology, Jeppiaar Nagar, Chennai, Tamil Nadu, India

## ABSTRACT

**Background:** The realization of computation-intensive applications such as real-time video processing, virtual/augmented reality, and face recognition becomes possible for mobile devices with the latest advances in communication technologies. This application requires complex computation for better user experience and real-time decision-making. However, the Internet of Things (IoT) and mobile devices have computational power and limited energy. Executing these computational-intensive tasks on edge devices may result in high energy consumption or high computation latency. In recent times, mobile edge computing (MEC) has been used and modernized for offloading this complex task. In MEC, IoT devices transmit their tasks to edge servers, which consecutively carry out faster computation.

**Methods:** However, several IoT devices and edge servers put an upper limit on executing concurrent tasks. Furthermore, implementing a smaller size task (1 KB) over an edge server leads to improved energy consumption. Thus, there is a need to have an optimum range for task offloading so that the energy consumption and response time will be minimal. The evolutionary algorithm is the best for resolving the multiobjective task. Energy, memory, and delay reduction together with the detection of the offloading task is the multiobjective to achieve. Therefore, this study presents an improved salp swarm algorithm-based Mobile Application Offloading Algorithm (ISSA-MAOA) technique for MEC.

**Results:** This technique harnesses the optimization capabilities of the improved salp swarm algorithm (ISSA) to intelligently allocate computing tasks between mobile devices and the cloud, aiming to concurrently minimize energy consumption, and memory usage, and reduce task completion delays. Through the proposed ISSA-MAOA, the study endeavors to contribute to the enhancement of mobile cloud computing (MCC) frameworks, providing a more efficient and sustainable solution for offloading tasks in mobile applications. The results of this research contribute to better resource management, improved user interactions, and enhanced efficiency in MCC environments.

# INTRODUCTION

Mobile cloud computing (MCC) is a substructure where both mobile tasks and data are kept and managed on great virtualized resources that are situated in the cloud

Corresponding author
Aishwarya R., aishwaryarajkumar20@outlook.com

infrastructure which can be accessed on request (*Zhou et al., 2023*). The computational severe tasks of mobile applications are uploaded to the cloud through a 4G or 5G connection to store and process, and then the outcomes are transported to the smart mobile devices (*Maray & Shuja, 2022*). Moreover, this procedure is called task offloading. So, MCC overwhelms the controlled resources of mobile devices and enhances both power utilization and response time. However, task offloading enhances the response time of delay-sensitive mobile users owing to the lower bandwidth and the higher potential of the internet (*Mohammed & Țăpuș, 2023*). Presently, there are many research works in cloud computing (CC) that goal to improve the computing abilities of resource-constrained mobile client devices by delivering mobile clients access to computing services, software, and cloud infrastructures (*Elgendy & Yadav, 2022*). For instance, Amazon web services are employed to defend and protect customer's personal information through their simple storage service (S3). Furthermore, numerous frameworks permit to procedure of data-intensive tasks distantly on cloud servers (*Verma, Tiwari & Hong, 2023*).

Offloading is the main specific feature of MCC and extends the implementation of the application among remote and local implementations (*Kim et al., 2024*). The selection may want to be adapted in response to variations in functioning situations like computational cost, total cost of execution, communication cost, response time, safety agent, and user input. Particularly, offloading is not constantly the greatest energy-efficient system, it takes additional energy when compared to the local implementation, mainly if the content is smaller (*Wu et al., 2024*). Offloading models are clustered into three wide classes such as virtualization, mobile agents, and client-server communication. Traditionally, computation offloading techniques have focused on single-site offloading that separates an application from a single remote server and mobile device. In contrast to single-site, multi-site computational offloading provides users with extra options. Many researchers have examined and constructed models that focus on multi-site offloading in current years (*Al-Hammadi et al., 2024*). The researchers showed that multi-site offloading may enhance the performance. As an outcome, it is considered as a normally accurate method. It covers more valued resources than a single site and utilizes less energy. In contrast, creating a selection for the multi-site offloading is nothing but a problem of NP-complete, and therefore defining the finest solution is a challenging task. Owing to the NP-complete feature of assigning numerous tasks to many sites of offloading, the common research works use meta-heuristics or heuristics to find a good result for scheduling issues than an optimal one (*Hosny et al., 2023*). Heuristics permit the scheduling issues solution more rapidly than the official one. Currently, evolution computing is employed to offload multi-site computations that determine a suitable solution for offloading on the mobile device in an adequate time (*Farahbakhsh, Shahidinejad & Ghobaei-Arani, 2023*). The main objective of evolution computing is to project computer systems for challenging complex search and optimizer issues by employing standards of natural assortment and genotype differences in nature.

The mobile edge computing (MEC) has emerged as a promising method for bringing edge servers closer to users while lowering latency and energy usage. However, the main problem is selecting whether to execute operations locally or offload them to MEC servers

while taking into account network circumstances, computational load, and resource availability. Inefficient offloading schemes can cause network congestion, increased expenses, and poor performance, needing clever optimization techniques to balance performance and resource constraints. The proposed method improved salp swarm algorithm-based Mobile Application Offloading Algorithm (ISSA-MAOA) is acceptable for mobile task offloading optimisation because of its low computing difficulty, robust global search capabilities, and suitability for dynamic and resource-constrained mobile MEC contexts. In contrast to other metaheuristic algorithms, salp swarm algorithm (SSA) successfully strikes a compromise between exploitation and examination, assuring real work distribution while evading local optima. Adaptive decision-making is made possible by its nature-inspired swarm intelligence system, which optimises work distribution to reduce latency and energy consumption.

The enhanced exploration-exploitation balance of the proposed technique ISSA-MAOA is the driving force behind its use for mobile task offloading optimisation. This balance is vital for managing the resource-constrained and dynamic nature of mobile MEC environments. For similar situations, a number of metaheuristic algorithms have been used, including genetic algorithms (GA), particle swarm optimization (PSO) and ant colony optimization (ACO); however, ISSA delivers more agility and convergence time. Though its competence is limited by its delayed convergence and early stagnation, the traditional SSA is renowned for its robust exploration ability. The enhanced ISSA integrates opposition-based learning, chaotic initialization, and adaptive weight methods to recover search competence and avoid entrapment in local optima. ISSA is therefore mainly well-suited for real-time mobile task offloading, where rapid decisions are vital to minimize latency, lower energy consumption, and exploit resource use in MEC systems.

This study presents an ISSA-MAOA technique for MEC. This technique harnesses the optimization capabilities of the ISSA to intelligently allocate computing tasks between mobile devices and the cloud, aiming to concurrently minimize energy consumption, memory usage, and reduce task completion delays. The proposed ISSA-MAOA aims to advance MCC frameworks, providing a more efficient and sustainable method for offloading tasks in mobile applications. The outcomes of this study lead to more effective resource allocation, smoother user interactions, and greater efficiency in MCC environment.

## LITERATURE REVIEW

In *Li et al. (2023)*, a novel multi-objective tactic dependent upon the biogeography-based optimizer (BBO) method is projected for MEC offloading to fulfil consumers' multiple desires (energy consumption, implementation time, and cost). In this tactic, a time-energy utilization and a cost techniques are built for task offloading initially. Dependent upon these techniques, the BBO system is presented for MEC to resolve the issue of multi-objective optimizer. In *Mahenge, Li & Sanga (2022)*, the main aim is to intend a task offloading system that reduces the complete energy utilization besides sustaining volume and delay requests. Therefore, we suggest a MEC-aided energy-effectual task offloading structure that influences the supportive MEC structure. To attain this, we offer a new

hybrid technique that is recognized as grey wolf optimization (GWO) and PSO to resolve the optimizer issue. In *Naouri et al. (2021)*, a three-layer task offloading structure termed DCC has been developed that contains three layers such as cloudlet, cloud, and device. In DCC, the tasks with higher calculating conditions are unloaded to the layers of cloudlet and cloud. Whereas the tasks with higher communication costs and lower computing were performed on the device layer, therefore DCC evades conveying huge volumes of data and can efficiently decrease the delay.

In *Liu et al. (2023)* proposed that the edge cloud computing (ECC) method be used to offload the calculations from smart mobile device (SMD) in order to reduce the total implementation cost of manifold devices. This method initially conveys the complete cost minimization issue below the restraints of application close target and connection period by reflecting the flexibility of SMD and edge cloud. Additionally, the computational resources of both the periphery cloud and SMD are restricted. In *Zhan et al. (2020)*, the issue of transferring results and resource allocation amongst manifold consumers is examined, with the assistance of a single baseline location, to achieve the optimal system-wide consumer efficacy, which is defined as a trade-off between energy utilization and task latency. In an optimizer, mobility is quantified. To confirm that the problem is NP-hard and to propose a heuristic mobility-aware offloading algorithm (HMAOA) for the purpose of obtaining the estimated optimal offloading structure. The distinctive global optimizer problem is transformed into numerous local optimizer issues. *Cui et al. (2022)* suggest a new offloading scheduling dependent upon multi-user fine-grained for IoT. By considering the energy consumption and delay, the computational offloading has been measured as a multi-objective optimization problem (CMOP), and an enhanced NSGA-II system is projected to resolve CMOP.

In *Bi et al. (2024)*, a cost-minimized computational offloading structure is expressed and resolved by a dual-phase optimizer algorithm called Lévy flight and Simulated Annealing-based GWO (LSAG). In the initial phase, the optimum edge range tactic is well-defined to deal with the situation of numerous accessible small base stations (SBS). *Pan et al. (2021)* project a multi-objective clustering evolution algorithm named multi-objective clustering evolutionary algorithm (MCEA) to diminish the energy consumption and cost of multi-flow implementation below the limit restraint. Initially, the sub-limit restraint is inserted at the time of initialization to produce extra early solutions that fulfil the limit restraint. Then, an adaptive clustering technique has been assumed to direct individuals to discover an appropriate mate at the time of the crossover process. At last, the prospects of crossover and change are vigorously attuned dependent upon the past data to manage the evolutionary track and speed of convergence. *Aishwarya & Mathivanan (2024)* proposed a structure setup to enhance consumption of power and performance in mobile environs by exploiting cloud computing by offloading computational tasks effectively from end device to cloud servers. *Aishwarya & Mathivanan (2023)* provides the diverse obstacles that mobile cloud offloading (MCO) faces, including platform variety, mobile cloud application security and privacy, defect tolerance, on-going connectivity, automatic offloading mechanisms, and offloading

economics and costs. The current research deficits and opportunities for future work are the subject of the open research issue in the MCO section. The most recent developments in MCO, such as the growing interest in edge offloading and the necessity of energy-efficient offloading techniques, are underscored by the most recent trends and research problems in the MCO method.

## THE PROPOSED METHOD

In this article, we have presented a new ISSA-MAOA model for MEC. This technique harnesses the optimization capabilities of the ISSA to intelligently allocate computing tasks between mobile devices and the cloud, aiming to concurrently minimize energy consumption, and memory usage, and reduce task completion delays. Figure 1 signifies the general architecture of the proposed ISSA-MAOA paradigm for MEC. The use of an ISSA to dynamically optimise task offloading decisions is one of the model's innovative features. The architecture represents the interaction of mobile devices, the MEC server, and the evolutionary optimisation process, which is reliable with the optional technique. The graphic shows how activities are categorised according to their computing difficulty, which is followed by an adaptive decision-making system that balances energy efficiency, reaction time, and computational load. The ISSA component is in charge of optimising resource allocation and preservation that task execution is attained professionally across mobile devices and cloud resources. This organised method advances overall system performance by lowering latency and increasing energy utilisation, making it a big step forward in MEC job offloading approaches.

The architecture consists of multiple key components:

1. MEC edge server & base station (BS): The edge servers communicate periodically with mobile devices to keep track of their computational status and available resources.

2. Mobile devices & IoT devices: These devices generate various computational tasks, ranging from simple to complex, requiring different levels of processing power.

3. Task offloading process:

   - The mobile device senses the input, estimates the computational complexity, and determines whether a task should be executed locally or offloaded.
   - The MEC server processes the task by considering factors such as energy consumption, memory availability, and network conditions.
   - The evolutionary algorithm (ISSA) is applied to optimize the task allocation decision.

4. ISSA optimization module: The ISSA is used to make optimal offloading decisions by balancing energy consumption, response time, and memory utilization.

The goal of the ISSA-MAOA technique is to enhance task execution efficiency, reduce delays, and optimize energy consumption, making it a more sustainable approach for mobile applications in MEC environments.

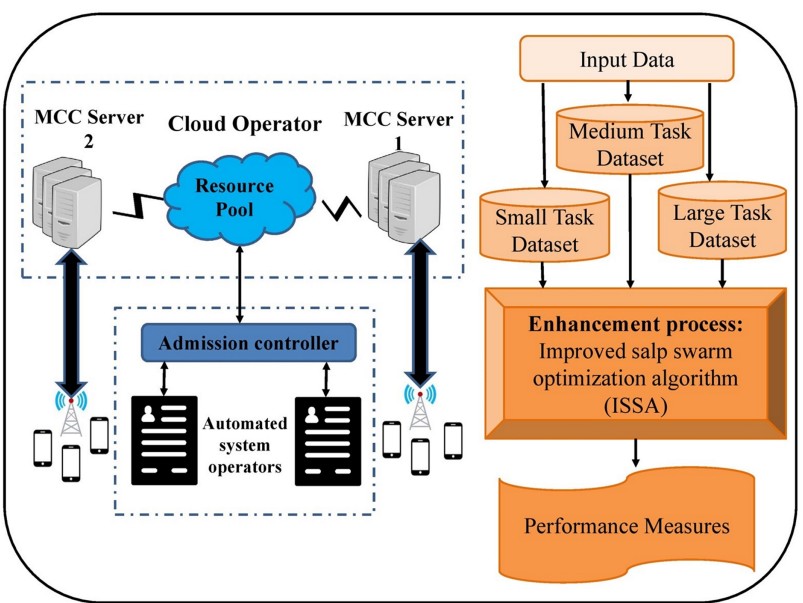

**Figure 1  Overall process of ISSA-MAOA technique.**

## System model

The MEC edge server refers to multiple IoT through BS. Periodic messages are swapped to maintain the MEC server updated with mobile devices (*Abbas et al., 2021*). This enables the MEC server to have perfect knowledge of interconnected devices, and their parameters, *viz*, $F_m$, $P_t$ and accessible bandwidth $B$.

In effective lifetime, IoT or mobile devices can impellent different tasks that ranges from face detection to smart transport systems. Some challenges require less CPU cycle and are simple for execution, while others need wide-ranging computations. When the device is generated to execute a specific task, then senses the input; computes $d$, $C$ to calculate that task and $\tau$ for delay-sensitive task. The MEC server used to direct the pre-fetched ($F_m$, $P_t$, and $B$) and received ($d$, $C$, and $\tau$ parameters to the MEC controller that consecutively perform the evolutionary algorithm to make offloading decisions.

## Algorithmic design of ISSA

The SSA is a nature-stimulated optimizer technique that exhibits the combined behavior of salps and sea invertebrates recognized for their swarm pattern (*Mohammed et al., 2024*). SSA is intended to challenge optimizer issues by imitating the social contacts and co-operation experimental in salp groups. This algorithm uses mathematical formulas to rule the decision-making and effect procedures of effective salps. The proposed method ISSA-MAOA's balanced exploration-exploitation mechanism aims to improve convergence time without resorting to premature optimization, a major drawback of conventional metaheuristic algorithms. ISSA-MAOA adds adaptive weight methods, chaotic mapping, and opposition-based learning to maintain an ideal balance between exploration and exploitation, in contrast to regular SSA, which may experience early

stagnation because of its biassed exploration or delayed convergence. ISSA-MAOA may examine a larger solution space in the early iterations and essence on refining high-quality solutions in later stages because to the adaptive weight mechanism, which dynamically modifies the search intensity based on the evolution stage. Furthermore, a varied population of solutions is guaranteed *via* chaotic initialization, which keeps the algorithm from being stuck in local optima. ISSA-MAOA is further strengthened by opposition-based learning, which speeds up convergence without sacrificing accuracy by assessing alternate solutions. ISSA-MAOA offers a more dependable, flexible, and real solution for resolving multi-objective optimization issues in dynamic MEC environments than the current SSA, PSO, ACO, and GA approaches.

The SSA includes the upgrade of every salp location that dependent upon its present location, the finest solution met so far, and the cooperative effect of the complete group. The location upgrade can be represented as in Eq. (1):

$$y_{1,j}^{t+1} = \begin{cases} F_j + w_1\left(\left(ub_j - lb_j\right)w_2 + lb_j\right) & w_3 \geq 0.5 \\ F_j - w_1\left(\left(ub_j - lb_j\right)w_2 + lb_j\right) & w_3 < 0.5 \end{cases} \tag{1}$$

$y_{1,j}^{t+1}$ and $F_j$ specifies the coordinates of the 1st salp location and food source position in the $j$th size, correspondingly. The lower and upper boundaries are signified by $lb_j$ and $ub_j$, correspondingly. The lower and upper boundaries are signified by $lb_j$ and $ub_j$, correspondingly. The dual measures $w_2$ and $w_3$ are randomly generated in the range of [0 *and* 1]. $w_1$ is employed to get constancy in the procedure of exploration and exploitation, demonstrating a crucial point parameter.

$$w_1 = 2e^{\left(\frac{-4t}{T}\right)^2}, \tag{2}$$

where, $T$ denotes the maximum iterations count and $t$ represents the current iteration. Equation (1) which was mentioned before, relates completely to the upgrading of the foremost salp location. To define the location upgrade to the follower salps, Eq. (3) is employed and delivered below.

$$y_{i,j}^{t+1} = \frac{1}{2}\left(y_{i,j}^t + y_{i-1,j}^t\right), \tag{3}$$

$y_{i,j}^t$ and $y_{i-1,j}^t$ represents the position of the $i$th follower in the $j$th dimension. The Eq. (3) based on Eq. (4) (*i.e.*, Newton's law of movement):

$$y_{i,j}^t = \frac{1}{2}a \times time^2 + u_0 \times time. \tag{4}$$

Here, *time* represents the time for each iteration, $y_{i,j}^t$ denotes the position of the $i$th follower in the $j$th dimension and $u_0$ represents the early speed, and $a$ is intended as:

$$a = \frac{u_{final}}{u_0} \tag{5}$$

whereas, $u_{final}$ denotes the final speed. $u_0$ signifies the early speed. It is important to remind that the follower and leader salp locations were upgraded by employing Eqs. (1) and (3). The followers are planned to imitate the leader movements till the highest amount of

iteration is attained, where the leader is upgraded liable on the accessibility of food source. At the time of iteration, the exploratory stage offers way to the exploitative stage in the parameter $w_1$.

## Locally weighted approach

The locally weighted (LW) approach is also recognized as a local search. It is a heuristic technique employed to discover the result of difficult optimizer problems. It contains nonstop altering of the current with adjacent solutions in the searching space. An effective local search is to discover an effectual method to pick suitable neighbors. The developed algorithm named local search algorithm (LSA), employs this local searching approach to progress the existing result, or salp, after entire iterations of the optimizer procedure.

Firstly, in each iteration $t$, a populace $pop^t$ with the dimension of $Npop$ salps and $y_i^t = \left( y_{i,1}^t, y_{i,2}^t, \ldots y_{i,dim}^t \right)$ will be enhanced by SSA as per the projected model and become $xnew_i^t$; Then, this salp is upgraded by the LW to become $ynew_i^t$ by utilizing this formulation:

$$weight_j = \frac{1}{\left( 1 + \exp\left( xnew_i^t - y_{i,j}^t \right) \right)} \tag{6}$$

$$ynew_i^t = xnew_{ii}^t + Z \times \left( weight_j \times \left( y_{r_1,j}^t - y_{r_2,j}^t \right) \right) \tag{7}$$

where $i = 1, 2, \ldots Npop$, $y_{r_1,j}^t$, $y_{r_2,j}^t$ are dual particles selected at random from the population $pop^t$ (except the existing particle $y_i^t$). Furthermore, $Z$ is randomly produced numbers by the magenta model on Levy distribution:

$$z = 0.01 \times \frac{b}{|q|^{\frac{1}{\infty}}}. \tag{8}$$

Here, $b$ and $q$ are strained from usual distributions as $b \sim N\left(0, \beta^2\right)$ and $q \sim N\left(0, \beta^2\right)$, and $\beta$ is produced by the given calculation:

$$\beta = \sqrt[\alpha]{\frac{\Gamma(1 + \infty) \sin\left(\frac{\pi\alpha}{2}\right)}{\Gamma\left\lceil\frac{\alpha+1}{2}\right\rceil \alpha 2^{\frac{\alpha-1}{2}}}} \tag{9}$$

whereas $\alpha$ denotes the stability index $\alpha \in [0, 2]$, which is also denoted as the Levy index.

## Upgrade salp followers' location

Owing to the advantages, the SSA enjoys are its flexibility and simplicity of execution that outcome from its direct mathematical formulation and use less parameters. The search efficacy for the food location is reduced over nonstop iteration of upgrading the leader position outcomes in an inactivity procedure. Moreover, in the SSA technique, the line among exploitation as well as exploration is not openly definite. The local stagnation issue is taken into consideration in this work; so we developed a change to the formulation for upgrading follower's locations in Eq. (5) in a salp chain over a mutation factor as:

$$ynew_i = y_i^t + rand \times mu \times \left(y_{r_1}^t - y_{r_2}^t\right) \tag{10}$$

where rand represents the randomly generated value within the range of $(0 \; and \; 1)$, $mu$ denotes the mutation factor set equivalent to 0.5, and $y_{r_1}^t$ and $y_{r_2}^t$ are dual random positions within the populace that dissimilar from position $y_j^t$.

The $y_j^t$ can connect with $y_{r_1}^t$ and $y_{r_2}^t$ regarding the location of other slaps. If there is a huge gap among $y_{r_1}^t$ and $y_{r_2}^t$, the individual getting the upgrade is probable to be anywhere in the center of that interval. In the exploration procedure, this kind of behavior is useful. When the space among $y_{r_1}^t$ and $y_{r_2}^t$ is smaller, the individual is more probable to look closely for likely solutions. If the $mu$ standard has been used, individuals will lead to the finest possible outcomes, and dissimilarity slowly declines as an outcome. The groups can efficiently examine and find potential places instead of depending only on the struggles. Simultaneously, rand is the randomly produced number from the range $[0 \; and \; 1]$ which is combined to change the food source location and take benefit of characteristic arbitrariness of SSA Eq. (1), effectively evading the local optimal issue, and allowing the convergence procedure to be restarted. Figure 2 depicts the block diagram of ISSA.

The configuration space includes decision variables that indicate the algorithm's decisions, such as whether a task is offloaded to an edge server or executed locally, the number of computational resources allocated, and the task scheduling order. The search space is defined by the range of feasible values for these variables, which might be binary (0 or 1) for offloading decisions or continuous (0 to 1) for resource allocation probabilities. The encoding type is critical in determining how solutions are encoded, with common options including binary encoding, continuous encoding, and hybrid encoding, which mixes the two to reflect offloading decisions and resource allocation at once. The fitness function assesses the quality of a solution by combining various objectives, such as minimizing job completion time, energy consumption, and cost while maximizing resource utilization, and frequently employs a weighted sum method. Constraints such as task deadlines, resource capacity limits, energy consumption thresholds, and network latency bounds are incorporated into the fitness function *via* penalty functions that penalize infeasible solutions, or handled *via* repair mechanisms that modify solutions to meet constraints.

## Process involved in ISSA-based offloading process

The optimal selection results in minimum energy consumption and response time of IoT devices that mathematically can be represented as in Eq. (11):

$$T_m > T_{0ff} \tag{11}$$
$$E_m > E_{0ff} \tag{12}$$

where $T_{off}$ and $T_m$ denotes the time that locally calculate the task at MEC servers and mobile devices, correspondingly. $E_{off}$ and $E_m$ are the energy depletion to locally perform tasks at the mobile devices and edge server, correspondingly. If $T_m$ and $E_m$ are lesser than $T_{off}$ and $E_{off}$ then it is more efficient to locally execute a task on mobile devices,

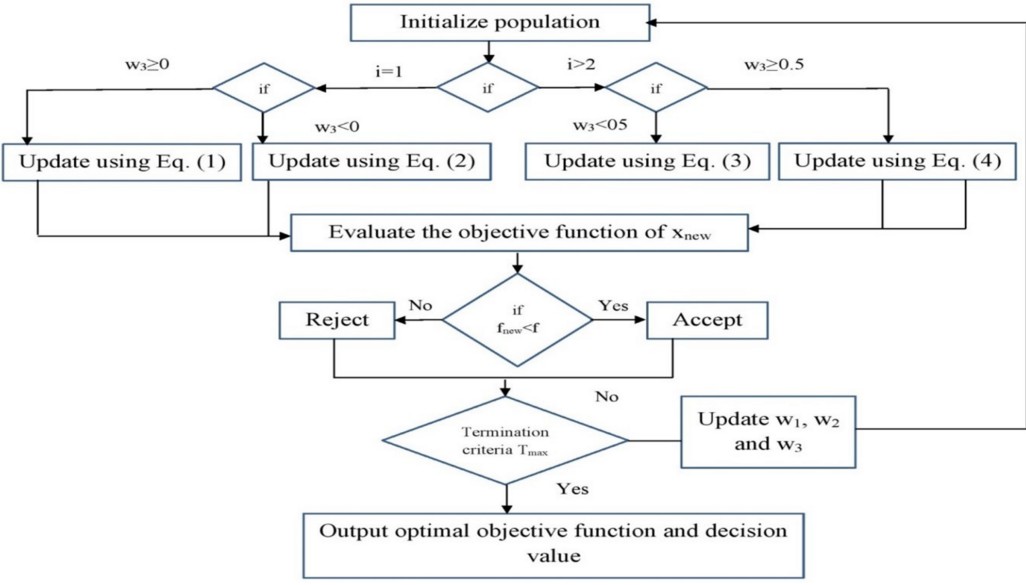

**Figure 2 Steps involved in ISSA.**

correspondingly. This is based on the time and quantity of energy needed to execute the task. The mobile device transmits the parameters including task size, network condition, and node capability to the MEC server. The controller tracks an evolutionary mechanism and chooses the task to be locally performed on the mobile device or sent for the computational of task offload. If $T_m$ is greater than $T_{off}$ and $E_m$ is greater than $E_{off}$ then the task offloading is only beneficial.

$$T_m = \frac{dC}{F_m} \tag{13}$$

$$T_{off} = \frac{d}{B} + \frac{dC}{F_s} + T_{wait} \tag{14}$$

$$E_m = \alpha CdF_m^2 \tag{15}$$

$$E_{off} = P_{t^*}\frac{d}{B} + E_{idle}. \tag{16}$$

$T_{off}$ involves the wait time $T_{wait}$ at server and transmission time for transferring information to the server. $d$ is the amount of data to be transmitted. $C$ is the number of CPU cycles required per bit of data. $Em$ represents the energy consumption for local computation. $\alpha$ is a hardware-dependent parameter related to the energy efficiency of the processor. When "$d$" is smaller and big number of "$C$" is taken for the task execution, then it is more efficient to offloading the task at MEC servers. Thus, if the "$d$" larger and smaller number of "$C$" takes adequate time to offload it to the server, then it is more efficient to locally execute a task on the device. Likewise, a heavily overloaded server having several procedures waiting in the buffer powers to execute the task locally which considers the data size and computation cycle.

Some IoT have sufficient computation resources to perform the task within a time. But, this decrease in $T_m$ take place at the cost of further draining of battery energy since $E_m$ is

| Algorithm 1   Pseudocode of ISSA. |
| --- |
| Every searching agent's dimension (D), upper bound (*ub*) and lower bound (*lb*), the fitness function (Fitness) and maximum iteration count (T) are inputs. |
| Cost function (Food Fitness) and optimum individual (Food Location) are the last outcomes. |
| As per population size *N*, *ub*, *lb*, Dimension *D*, initial Salp populace. |
| As per to the fitness function, pick the smallest expensive individual in the populace as the Food Fitness. |
| while (end condition does not hold) |
| calculate *w*1 by Eq. (2) |
| for (all Salp (Salp position)) |
| if ($i \leq N/2$) then |
| Upgrade the position of the Salp leaders by Eq. (1) |
| else |
| Upgrade the position of the follower's Salp by Eq. (10) |
| if (random < 0.5) |
| Utilize (LW) model as an Algorithm |
| Calculate the fitness value of $ynew_i$ and specified it as NewFitness |
| if (NewFitness < *Food* Fitness) then |
| Upgrade the Food position and Food Fitness |
| end |
| end |

directly relative to the CPU cycle. $\alpha$ refers to the architectural constant of the CPU. $E_{off}$ denotes energy utilization during the offloading task. Also, it involves energy dissipation once the device is idle while executing offloading task. If the network condition is unfavorable, then it is more efficient for the task execution on a mobile device. The MEC server is used to receive the parameter from each node in responsibility and make wise decisions on network-. Note that the server enhances its efficiency and decides for a single node rather it intends to take accumulative decisions and enhance the network performance.

Consider that the MEC server has complete knowledge of $P_t$, $F_m$, $\omega$, and $B$. The server should make optimum decisions and select the task for the offloading task computation so that latency and energy consumption can be reduced. Since saving the node's energy and reducing the response time are minimization nature problems,

$$f(T) = \sum_{i=1}^{k}\left(\frac{d_i}{B_i} + \frac{d_i C_i}{F_s}\right) + \sum_{j=1}^{l}\frac{d_j C_j}{F_{m,j}} \forall = k + l \tag{17}$$

$$f(E) = \sum_{i=1}^{k}\left(P_{t,i^*}\frac{d_i}{B_i} + E_{idle,i}\right) + \sum_{j=1}^{l}\alpha C_j d_j F_m^2. \tag{18}$$

Equations (17) and (18) define the objective function for energy consumption and response time, correspondingly, where $f(E)$ and $f(T)$ are the objective functions for energy

and time, correspondingly. The rudimentary concept of optimization in this algorithm is used and develop the new design variant to resolve these problems using mathematical modelling.

The proposed approach shows advantages in work offloading and processing performance, which boosts the efficacy of IoT systems, smart healthcare, augmented reality, and remote-control applications. ISSA-MAOA is a valuable solution for next-generation computing environments because of these advancements, which allow for improved user experiences, reduced operational costs, and increased system stability. Even in unexpected MEC circumstances, ISSA guarantees optimal performance by constantly modifying task offloading options depending on shifting bandwidth, latency, and resource availability. IoT networks, mobile healthcare, and edge-based smart applications where network conditions are extremely dynamic and necessitate real-time optimization benefit greatly from its adaptability.

## EXPERIMENTAL VALIDATION

The proposed method ISSA-MAOA overcomes the drawbacks of the classic SSA, including poor solution variety and early convergence, by incorporating enhancements including adaptive parameter tuning, dynamic population management, and hybrid exploration-exploitation techniques. The experiments were carefully planned and carried out to assess the effectiveness, scalability, and resource usage of the ISSA-MAOA based approach. The hardware configuration comprised cloud servers housed on AWS EC2 instances, edge servers with Intel Xeon E5 CPUs and 128 GB of RAM, and mobile devices with Qualcomm Snapdragon 865 processors. The software environment included MATLAB R2022a and Python 3.9 for algorithm development and simulations, Android 11 for mobile devices, and Ubuntu 20.04 for edge/cloud nodes. NS-3 was used to simulate the network circumstances, enabling controlled changes in server loads, latency, and bandwidth. IoT application workloads from Google Edge AI benchmarks and simulated activities ranging from computationally demanding deep learning jobs (1 GB) to lightweight sensor data processing (1 KB) were among the real-world and synthetic datasets used in the project. Energy consumption, execution time, latency, task offloading success rate, and server utilization were the main valuation criteria. ISSA-MAOA was contrasted with the conventional SSA, PSO, GA, and ACO algorithms. The experimental design guaranteed a thorough and repeatable evaluation of ISSA-MAOA's adaptability and efficiency in MEC environment optimization.

The performance of the ISSA-MAOA technique was measured using important parameters such as task completion time, energy consumption, resource utilization, cost, scalability, and convergence rate. These standards were chosen to provide a full assessment of the proposed method's efficiency and efficacy. The ISSA-based method was compared to various other task offloading optimization algorithms, including the SSA, PSO, GA, greedy algorithms, and random offloading. The results showed that the ISSA-based method outperformed the baseline methods in the majority of scenarios, with important developments in task completion time, energy efficiency, resource utilization, and costs. Also, the ISSA-based technique established improved scalability and faster convergence,

making it appropriate for large-scale MEC applications. Statistical analysis and visualizations, such as graphs and tables, were employed to confirm the importance of the findings. Finally, the trials demonstrated the resilience and efficiency of the ISSA-based strategy for optimizing mobile task offloading, emphasizing its latent for real-world applications in MEC.

Minimal computation is typically required for smaller tasks, which are typically between 1 and 100 KB in size. These tasks can be executed locally on mobile devices with minimal impact on battery life and processing capacity. Offloading such minor duties to an edge server may result in unnecessary transmission delays, as the time required to transmit and receive results may exceed any computational advantages that may be obtained from edge processing. Moreover, the coincident offloading of a large number of small duties may result in network congestion and server overload, which reduces the overall capability of the system. Consequently, ISSA-MAOA prioritizes local execution for lesser tasks unless there is a substantial computational advantage in offloading. the task size plays a crucial role in decisive the efficiency of task offloading decisions. Smaller tasks are generally more suitable for local execution, while medium and large tasks benefit from intelligent offloading approaches to optimize performance. By dynamically adapting to task sizes and network conditions, ISSA-MAOA ensures well-organized resource utilization, reduced energy consumption, and improved execution speed, making it well-suited for real-world MEC applications. Offloading large tasks to MEC servers reduces processing time but may introduce network transmission delays and increased power consumption due to data transfers. Efficient task scheduling and adaptive offloading strategies, like ISSA-MAOA, help balance these trade-offs, optimizing energy efficiency, memory management, and real-time responsiveness.

In this section discussed a detailed result analysis of the ISSA-MAOA technique. In Table 1, a comprehensive energy consumption (ECON) result of the ISSA-MAOA technique is compared (*Abbas et al., 2021*). In Fig. 3, the comparative ECON results of the ISSA-MAOA technique with recent models under small tasks (1 kb) are provided. The figure stated that the ACO algorithm reaches worse results with increased ECON values. Though the GWO and WOA models obtain slightly decreased ECON values, the ISSA-MAOA technique reaches better performance with minimal ECON values of 6.93, 5.51, 9.86, 8.70, and 13.22 mJ under 20–100 tasks, correspondingly.

In Fig. 4, the comparative ECON outcomes of the ISSA-MAOA approach with the current method below medium tasks (500 kb) are delivered. The figure specified that the ACO algorithm reaches worse outcomes with improved ECON values. Though the GWO and WOA techniques attain slightly reduced ECON values, the ISSA-MAOA method reaches superior performance with the least ECON values of 8.94, 10.52, 12.48, 12.85, and 14.22 mJ under 20–100 tasks, respectively.

In Fig. 5, the comparative ECON outcomes of the ISSA-MAOA model with current techniques below large tasks (1,000 kb) are provided. The figure specified that the ACO algorithm reaches worse outcomes with amplified ECON values. Though the GWO and WOA approaches gain slightly reduced ECON values, the ISSA-MAOA system reaches

**Table 1 ECON analysis of ISSA-MAOA technique with other methods under various tasks.**

Energy consumption (module)

| Number of tasks | ACO protocol | GWO protocol | WOA protocol | ISSA-MAOA |
|---|---|---|---|---|
| **Small task (1 kb)** | | | | |
| 20 | 32.69 | 11.38 | 15.51 | 6.93 |
| 40 | 36.95 | 11.51 | 17.06 | 5.51 |
| 60 | 36.95 | 12.67 | 19.00 | 9.86 |
| 80 | 39.15 | 13.71 | 20.94 | 8.70 |
| 100 | 42.12 | 15.64 | 22.88 | 13.22 |
| **Medium task (500 kb)** | | | | |
| 20 | 34.51 | 13.25 | 18.41 | 8.94 |
| 40 | 37.21 | 13.99 | 22.10 | 10.52 |
| 60 | 40.65 | 15.96 | 22.96 | 12.48 |
| 80 | 42.49 | 18.29 | 24.92 | 12.85 |
| 100 | 45.32 | 20.50 | 27.50 | 14.22 |
| **Large task (1,000 kb)** | | | | |
| 20 | 35.80 | 14.37 | 21.89 | 9.17 |
| 40 | 36.41 | 14.87 | 22.62 | 8.08 |
| 60 | 42.57 | 17.33 | 24.96 | 12.50 |
| 80 | 44.05 | 19.55 | 26.19 | 14.36 |
| 100 | 48.11 | 22.75 | 28.53 | 16.56 |

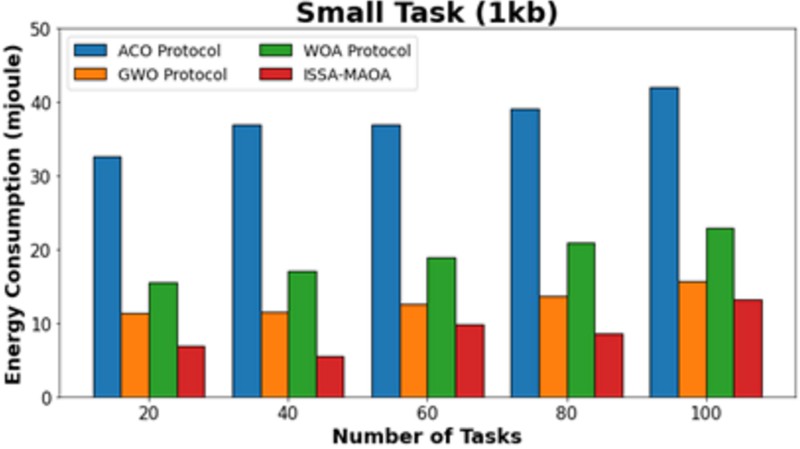

**Figure 3 ECON analysis of ISSA-MAOA technique under small tasks (1 kb).**

superior performance with the least ECON values of 9.17, 8.08, 12.50, 14.36, and 16.56 mJ under 20–100 tasks, respectively.

In Table 2, a comprehensive delay (DEL) outcome of the ISSA-MAOA technique is compared. In Fig. 6, the comparative DEL results of the ISSA-MAOA system with current techniques under small tasks (1 kb) are delivered. The figure specified that the ACO system

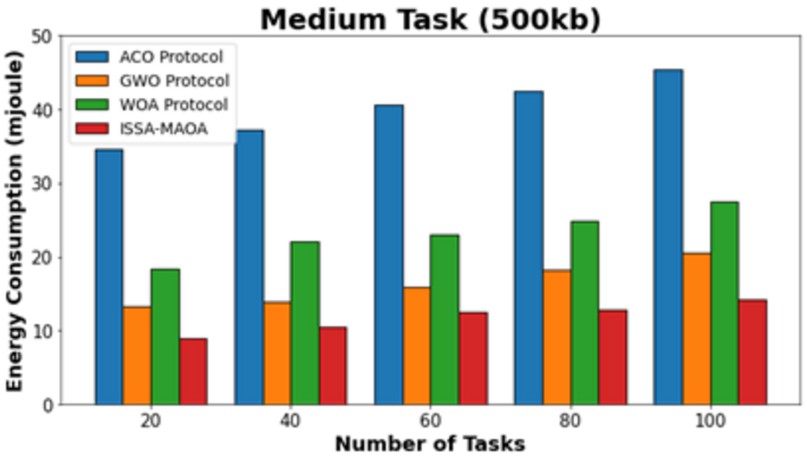

**Figure 4 ECON analysis of ISSA-MAOA technique under medium tasks (500 kb).**

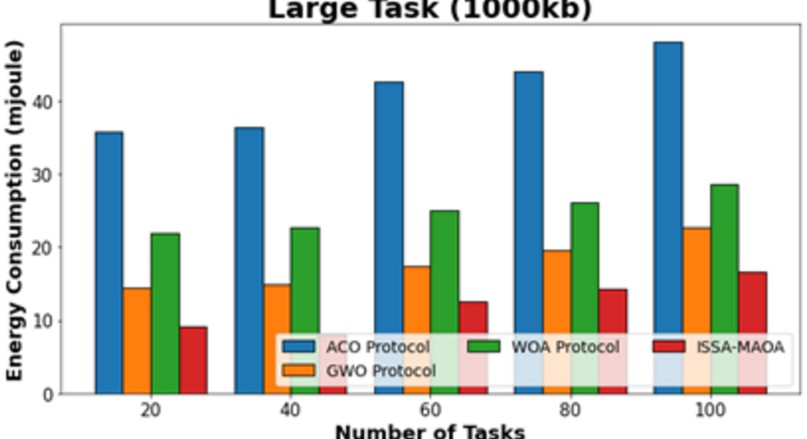

**Figure 5 ECON analysis of ISSA-MAOA technique under large tasks (1,000 kb).**

attains worse results with amplified DEL values. Though the GWO and WOA approaches got slightly reduced DEL values, the ISSA-MAOA method reaches superior performance with least DEL values of 31.73, 36.81, 32.73, 38.40, and 36.27 ms under 20–100 tasks, respectively.

In Fig. 7, the comparative DEL results of the ISSA-MAOA method with recent techniques under medium tasks (500 kb) are provided. The figure stated that the ACO algorithm extents worse outcomes with amplified DEL values. Though the GWO and WOA systems got slightly declined DEL values, the ISSA-MAOA model reaches better performance with minimal DEL values of 35.72, 38.34, 30.75, 36.96, and 29.08 ms under 20–100 tasks, correspondingly.

In Fig. 8, the comparative DEL outcomes of the ISSA-MAOA system with recent approaches below large tasks (1,000 kb) are provided. The figure definite that the ACO

**Table 2 DEL analysis of ISSA-MAOA technique with other methods under various tasks.**

**Delay (ms)**

| Number of tasks | ACO protocol | GWO protocol | WOA protocol | ISSA-MAOA |
|---|---|---|---|---|
| **Small task (1 kb)** | | | | |
| 20 | 57.77 | 40.08 | 52.10 | 31.73 |
| 40 | 60.04 | 41.67 | 52.33 | 36.81 |
| 60 | 61.18 | 43.03 | 55.51 | 32.73 |
| 80 | 62.31 | 42.58 | 54.60 | 38.40 |
| 100 | 65.71 | 42.80 | 57.32 | 36.27 |
| **Medium task (500 kb)** | | | | |
| 20 | 60.63 | 44.64 | 56.58 | 35.72 |
| 40 | 62.07 | 45.60 | 55.38 | 38.34 |
| 60 | 62.30 | 44.88 | 56.10 | 30.75 |
| 80 | 64.21 | 47.27 | 58.72 | 36.96 |
| 100 | 69.23 | 47.50 | 61.59 | 29.08 |
| **Large task (1,000 kb)** | | | | |
| 20 | 68.70 | 49.32 | 61.21 | 33.01 |
| 40 | 69.58 | 52.40 | 62.31 | 42.04 |
| 60 | 72.88 | 52.84 | 64.96 | 43.04 |
| 80 | 72.88 | 55.93 | 65.40 | 37.31 |
| 100 | 76.63 | 57.25 | 67.60 | 44.24 |

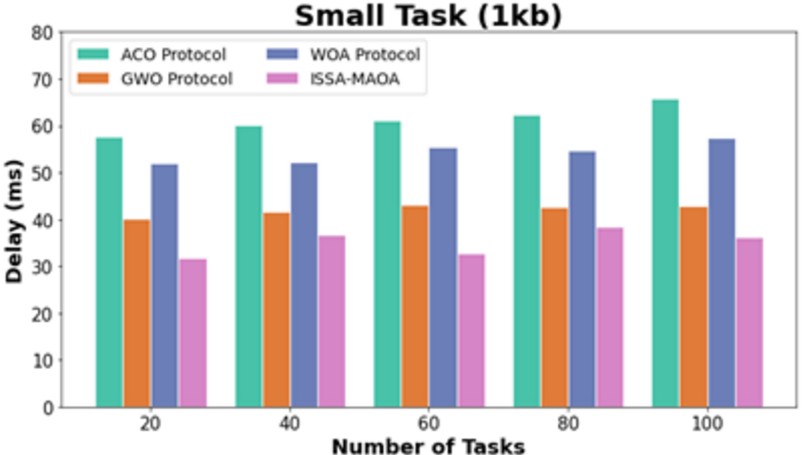

**Figure 6 DEL analysis of ISSA-MAOA technique under small tasks (1 kb).**

system reaches worse results with amplified DEL values. Though the GWO and WOA methodologies acquire slightly reduced DEL values, the ISSA-MAOA procedure reaches superior performance with marginal DEL values of 33.01, 42.04, 43.04, 37.31, and 44.24 ms under 20–100 tasks, correspondingly.

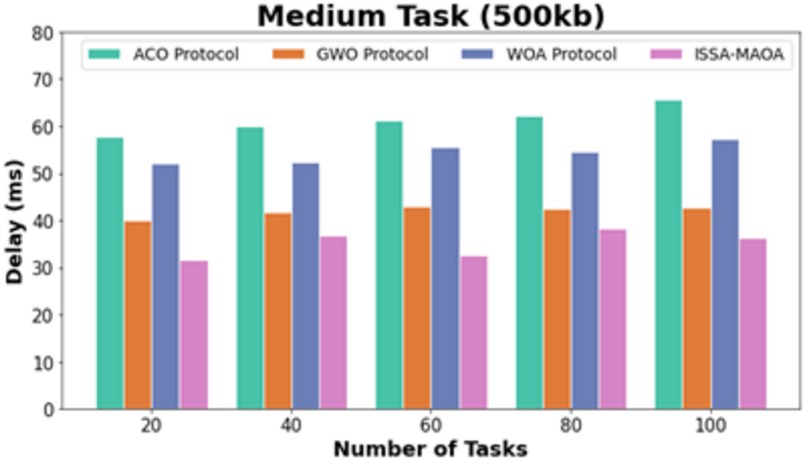

**Figure 7  DEL analysis of ISSA-MAOA technique under medium tasks (500 kb).**

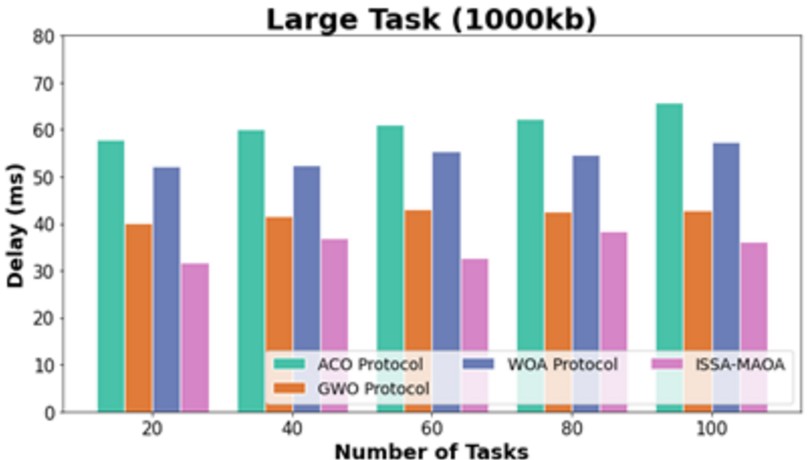

**Figure 8  DEL analysis of ISSA-MAOA technique under large tasks (1,000 kb).**

The number of offloading tasks (NOOT) outcomes of the ISSA-MAOA technique is compared with existing models in Table 3. Figure 9 exhibits a comparative NOOT result of the ISSA-MAOA system with current techniques on small tasks (1 kb). The results highlighted that the ISSA-MAOA approach reaches better performance with maximum NOOT values. With 20 tasks, the ISSA-MAOA technique gains a higher NOOT of 20.19 whereas the ACO, GWO, and WOA obtain lower NOOT of 11.38, 14.76, and 12.86, correspondingly. Meanwhile, with 100 tasks, the ISSA-MAOA method gains a greater NOOT of 77.09 whereas the ACO, GWO, and WOA get worse NOOT of 66.65, 70.03, and 71.93, correspondingly.

Figure 10 exhibits a comparative NOOT outcome of the ISSA-MAOA system with current techniques on medium tasks (500 kb). The results emphasized that the ISSA-MAOA system reaches enhanced performance with the greatest NOOT values. With

**Table 3 NOOT analysis of ISSA-MAOA model with recent methods below various tasks.**

**Number of offloading tasks**

| Number of tasks | ACO protocol | GWO protocol | WOA protocol | ISSA-MAOA |
|---|---|---|---|---|
| **Small task (1 kb)** | | | | |
| 20 | 11.38 | 14.76 | 12.86 | 20.19 |
| 40 | 20.24 | 28.47 | 23.20 | 32.58 |
| 60 | 40.49 | 46.40 | 42.81 | 49.46 |
| 80 | 53.57 | 60.54 | 59.06 | 64.49 |
| 100 | 66.65 | 70.03 | 71.93 | 77.09 |
| **Medium task (500 kb)** | | | | |
| 20 | 16.12 | 17.44 | 16.34 | 23.63 |
| 40 | 29.35 | 32.43 | 31.55 | 36.30 |
| 60 | 42.79 | 47.87 | 46.76 | 52.61 |
| 80 | 55.36 | 61.98 | 61.54 | 69.27 |
| 100 | 68.37 | 74.32 | 73.22 | 78.41 |
| **Large task (1,000 kb)** | | | | |
| 20 | 14.67 | 17.95 | 16.92 | 22.64 |
| 40 | 30.85 | 34.13 | 32.90 | 38.61 |
| 60 | 41.71 | 51.13 | 53.18 | 58.04 |
| 80 | 54.00 | 59.94 | 58.30 | 65.62 |
| 100 | 69.36 | 76.32 | 74.68 | 80.98 |

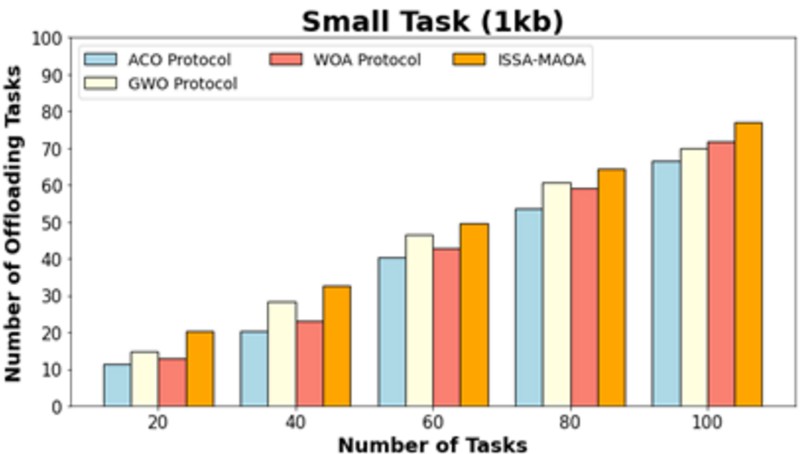

**Figure 9 NOOT analysis of ISSA-MAOA technique under small tasks (1 kb).**

20 tasks, the ISSA-MAOA procedure gains a greater NOOT of 23.63 while the ACO, GWO, and WOA gain worse NOOT of 16.12, 17.44, and 16.34, respectively. Meanwhile, with 100 tasks, the ISSA-MAOA method gains upper NOOT of 78.41 while the ACO, GWO, and WOA approaches got lower NOOT of 68.37, 74.32, and 73.22, respectively.

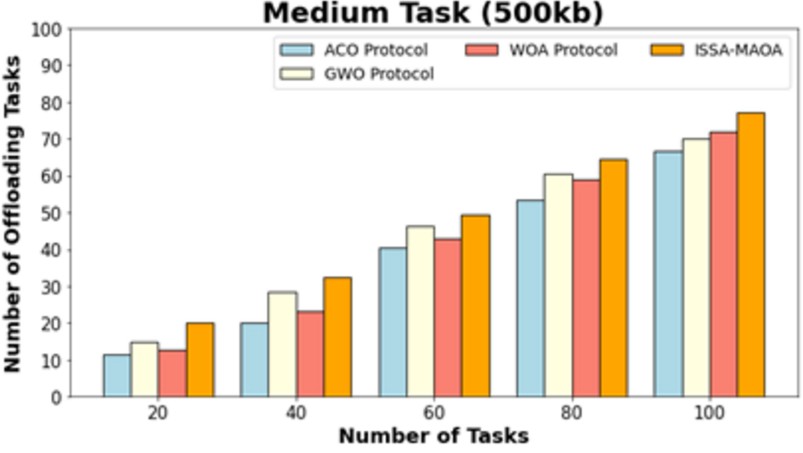

**Figure 10 NOOT analysis of ISSA-MAOA technique under medium tasks (500 kb).**

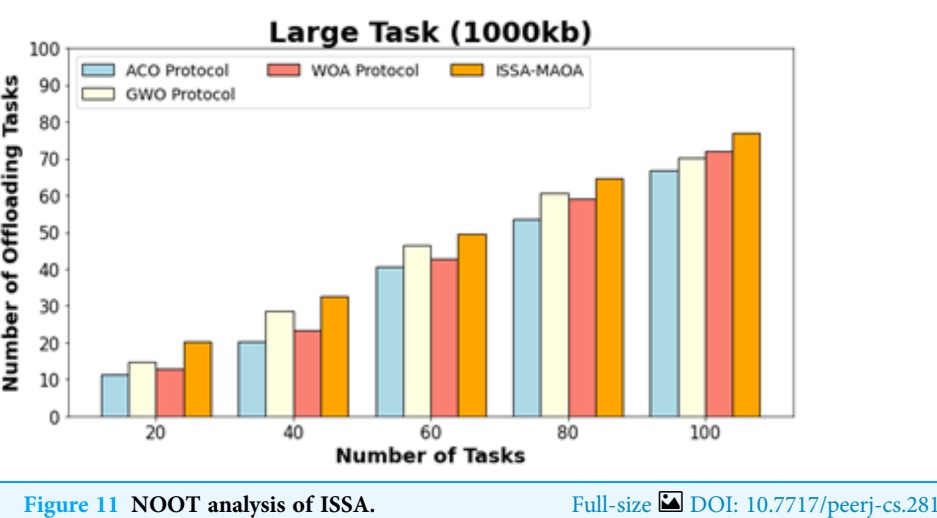

**Figure 11 NOOT analysis of ISSA.**

Figure 11 exhibits a comparative NOOT outcome of the ISSA-MAOA model with current models on superior tasks (1,000 kb). The results highlighted that the ISSA-MAOA system reaches superior performance with maximum NOOT values. With 20 tasks, the ISSA-MAOA model gains greater NOOT of 22.64 whereas the ACO, GWO, and WOA get lower NOOT of 14.67, 17.95, and 16.92, correspondingly. Meanwhile, with 100 tasks, the ISSA-MAOA methodology gains greater NOOT of 80.98 while the ACO, GWO, and WOA get lower NOOT of 69.36, 76.32, and 74.68, respectively.

The space-saving (SS) results of the ISSA-MAOA system are equated with current techniques in Table 4. Figure 12 displays a comparative SS outcome of the ISSA-MAOA system with current techniques on small tasks (1 kb). The results emphasized that the ISSA-MAOA model reaches superior performance with the largest SS values. With 20 tasks, the ISSA-MAOA system gains greater SS of 78.19% whereas the ACO, GWO, and WOA gets worse SS of 63.38%, 73.76%, and 71.86%, correspondingly. Meanwhile, with

**Table 4 SS analysis of ISSA-MAOA technique with recent methods under various tasks.**

Space savings (%)

| Number of tasks | ACO protocol | GWO protocol | WOA protocol | ISSA-MAOA |
|---|---|---|---|---|
| **Small task (1 kb)** | | | | |
| 20 | 63.38 | 73.76 | 71.86 | 78.19 |
| 40 | 71.24 | 82.47 | 77.20 | 87.58 |
| 60 | 40.49 | 46.40 | 42.81 | 50.46 |
| 80 | 67.57 | 79.54 | 70.06 | 75.49 |
| 100 | 76.65 | 91.03 | 90.93 | 94.09 |
| **Medium task (500 kb)** | | | | |
| 20 | 65.12 | 69.44 | 67.34 | 75.63 |
| 40 | 79.35 | 88.43 | 82.55 | 90.30 |
| 60 | 60.79 | 67.87 | 65.76 | 72.61 |
| 80 | 73.36 | 77.98 | 71.54 | 88.27 |
| 100 | 82.37 | 87.32 | 85.22 | 90.41 |
| **Large task (1,000 kb)** | | | | |
| 20 | 64.67 | 77.95 | 71.92 | 84.64 |
| 40 | 85.85 | 89.13 | 82.90 | 90.61 |
| 60 | 55.71 | 69.13 | 67.18 | 71.04 |
| 80 | 69.00 | 77.94 | 69.30 | 84.62 |
| 100 | 87.36 | 89.32 | 86.68 | 93.98 |

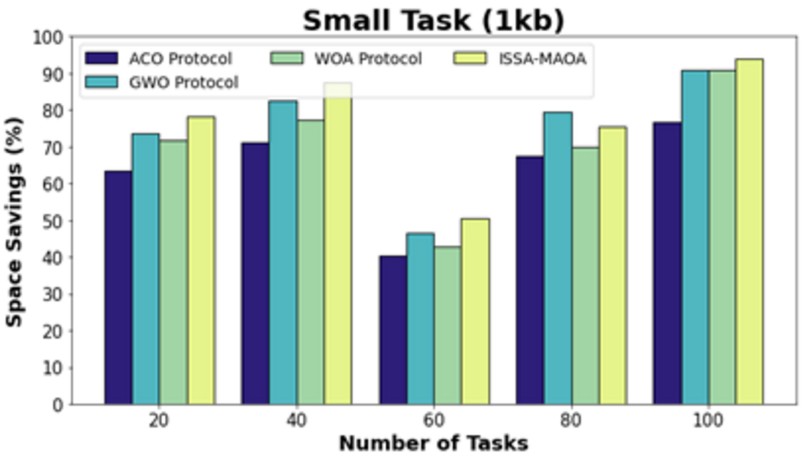

**Figure 12 SS analysis of ISSA-MA.**

100 tasks, the ISSA-MAOA system gains greater SS of 94.09% whereas the ACO, GWO, and WOA acquire lower SS of 76.65%, 91.03%, and 90.93%, correspondingly.

Figure 13 exhibits a comparative SS result of the ISSA-MAOA model with recent techniques on medium tasks (500 kb). The results underlined that the ISSA-MAOA system reaches superior performance with the highest SS values. With 20 tasks, the ISSA-MAOA model acquires higher SS of 75.63% while the ACO, GWO, and WOA models get lower SS

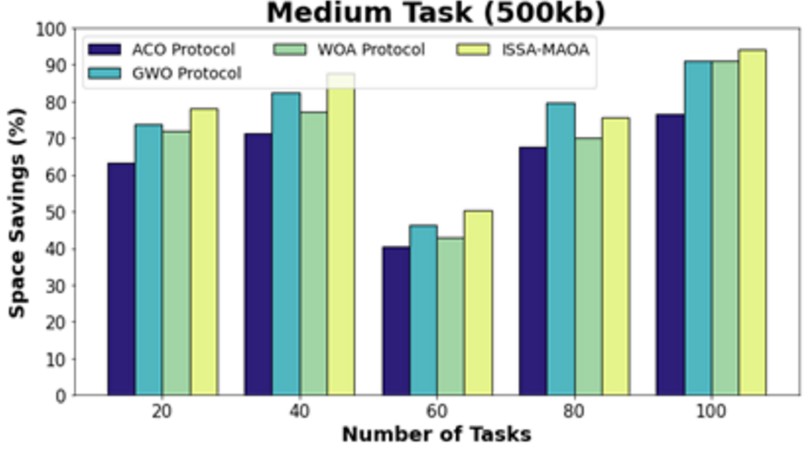

**Figure 13** SS analysis of ISSA-MA.

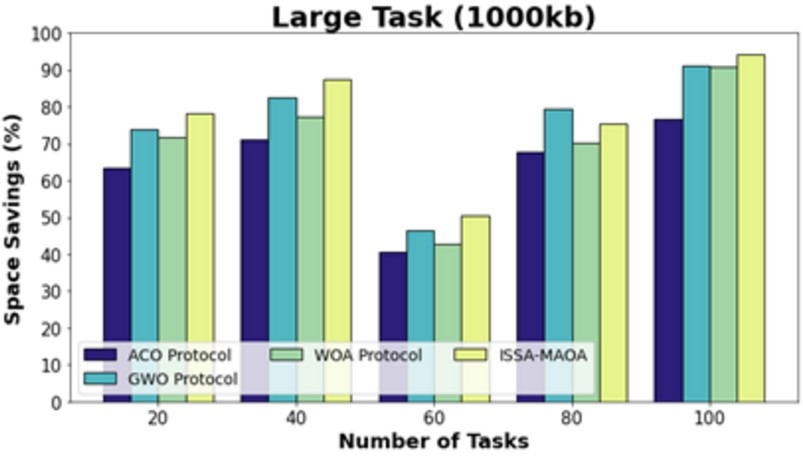

**Figure 14** SS analysis of ISSA-MA.

of 65.12%, 69.44%, and 67.34%, respectively. Meanwhile, with 100 tasks, the ISSA-MAOA method gains greater SS of 90.41% whereas the ACO, GWO, and WOA methodologies get lower SS of 82.37%, 87.32%, and 85.22%, correspondingly.

Figure 14 exhibits a comparative SS outcome of the ISSA-MAOA model with current approaches on larger tasks (1,000 kb). The results emphasized that the ISSA-MAOA model reaches superior performance with maximum SS values. With 20 tasks, the ISSA-MAOA method gains greater SS of 84.64% whereas the ACO, GWO, and WOA get lower SS of 64.67%, 77.95%, and 71.92%, correspondingly. Meanwhile, with 100 tasks, the ISSA-MAOA method gains upper SS of 93.98% while the ACO, GWO, and WOA gain lower SS of 87.36%, 89.32%, and 86.68%, respectively.

Therefore, the ISSA-MAOA technique can be found to be an effective approach for offloading process in the MCC environment.

## CONCLUSION

In this study presented a novel ISSA-MAOA method for MEC with optimization capabilities of the ISSA to intelligently allocate computing tasks between mobile devices and the cloud, aiming to concurrently minimize energy consumption, memory usage, and reduce task completion delays. The limitations of proposed method using in real-time applications like AR/VR or autonomous driving, ISSA's iterative nature may be too slow to meet millisecond-level latency demands. Similarly, in ultra-low-power IoT devices, its computational overhead may be too high for efficient execution. The proposed method ISSA-MAOA enhances mobile task offloading by optimizing resource allocation, reducing energy consumption, and minimizing execution delay. Its adaptive mechanisms ensure faster convergence, better load balancing, and improved task scheduling in dynamic edge computing environments. The ISSA-MAOA has significant implications for mobile task offloading by increasing computational resource management, reducing latency, and enhancing energy capability in edge computing environments. It is the optimal choice for IoT, 5G, and cloud-edge applications due to its adaptive optimization, which assurances the seamless distribution of workloads. ISSA contributes to the advancement of mobile computing systems that are more efficient, intelligent, and rapid by enabling real-time decision-making and optimizing multi-objective trade-offs.

Future work can focus on optimizing ISSA for applications like autonomous vehicles, AR/VR, and real-time gaming, where strict latency restraints require faster convergence and lightweight execution. In addition, the integration of multi-objective optimization can further balance the trade-offs between energy consumption, latency, and cost in heterogeneous periphery computing environments.

### Funding
The authors received no funding for this work.

### Competing Interests
The authors declare that they have no competing interests.

### Author Contributions
- Aishwarya R. conceived and designed the experiments, performed the computation work, prepared figures and/or tables, and approved the final draft.
- Mathivanan G. performed the experiments, analyzed the data, authored or reviewed drafts of the article, and approved the final draft.

### Data Availability
  The code and datasets are available in the Supplemental Files.

## Supplemental Information

Supplemental information for this article can be found online at http://dx.doi.org/10.7717/peerj-cs.2818#supplemental-information.

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
