# Peer review of "Improved salp swarm algorithm based optimization of mobile task offloading"

_PeerJ Computer Science, doi:10.7717/peerj-cs.2818_

## Round 0.1 · original submission · Major Revisions

Dear authors,


Reviewers have now commented on your article. We do encourage you to address the concerns and criticisms of the reviewers with respect to reporting, experimental design, and validity of the findings and resubmit your article once you have updated it accordingly. Following should also be addressed:

1. A major criticism for the current version of the paper is the lack of justification for the use of the salp swarm algorithm as the optimization method for the model. The motivation and reason of using salp swarm algorithm among many other metaheuristic algorithms for the focused problem should be mentioned. Therefore, you need to add a standard evolutionary algorithm as an additional point of comparison for your algorithm.
2. Configuration space of salp swarm algorithm should be detailed. It should be more specific and comprehensive. Representation scheme (encoding type) and fitness function with constraint functions should be clearly provided. How constraints (for example: for decision variables) are handled should also be provided.
3. Please pay special attention on the usage of abbreviations. Spell out the full term at its first mention, indicate its abbreviation in parenthesis and use the abbreviation from then on.
4. Please pay attention on the usage of blank characters.
5. Equations should be used with correct equation number. Please do not use “as follows”, “given as”, etc. Explanation of the equations should also be checked. All variables should be written in italic as in the equations. Their definitions and boundaries should be defined. Necessary references should be provided.
6. Many of the equations are part of the related sentences. Attention is needed for correct sentence formation.
7. All of the values for the parameters of all algorithms should be given.
8. Line between 103-111 should be re-written by not using “we”. According to the references, “you” did not perform these studies.
9. What does “d” represent in Equation (16) is not clear. What is input dataset?
10. There are major English grammar and writing style errors that should be corrected. For example: “The authors [22] provides…”, “…energy consumption and latency, computational offloading has been…”. The paper strictly needs proofreading.
11. It is recommended that the paper's experimental results be discussed in greater depth, with additional recommendations and conclusions provided. The conclusion section is lacking in several respects. Firstly, it is essential to describe the academic implications, main findings, shortcomings and directions for future research. Secondly, the conclusion is currently confusing. It is necessary to clarify what will happen next and what we should expect from future papers. To address these issues, the conclusion should be rewritten, taking the following comments into consideration:

- Highlight your analysis and reflect only the important points for the whole paper.
- Mention the benefits
- Mention the implication in the last of this section.

Best wishes,

·

Basic reporting

• The article provides a clear background on the problem of computation-intensive tasks in mobile devices and the challenges associated with energy consumption and latency.
• The proposed ISSA-MAOA method is explained with its goal to optimize energy consumption, memory usage, and delay in mobile edge computing (MEC).
• The article provide more references to recent studies or related work in the field of mobile edge computing and optimization techniques, which could enhance the context and credibility.
• Figures, such as Fig. 1 components should described in detail of their novelty, also their role in the methodology is unclear. A better explanation of the figure would help readers understand the procedure visually.
• Include a section that thoroughly discusses optimization techniques such as the Salp Swarm Algorithm (SSA) to highlight the novelty of the ISSA-MAOA.
• Expand on Fig. 1, describing its novel components and how it aligns with the proposed method.
"• Replace "ISSAís" with "ISSA's" for correct use of possessive form.
• Example: "Execution time and latency key factors in real-time applications were reduced..." should read:
"Execution time and latency, key factors in real-time applications, were reduced..."
• Avoid repetition. For instance:
"ISSA consistently beat other benchmark algorithms" and "ISSA achieved a 96% offloading success rate, surpassing SSA, PSO, and GA" can be streamlined to avoid redundancy.

Experimental design

• The study focuses on solving a multi-objective optimization problem using an evolutionary algorithm, which is a well-established approach for such issues.
• The ISSA-MAOA approach is presented as a novel technique, which likely addresses the limitations of traditional task offloading strategies
• The methodology lacks clarity on how the experiments were designed and conducted, including details such as the dataset used, metrics for evaluation, and specific comparisons made with existing methods.
• There is no clear explanation of how the size of the tasks (e.g., "smaller size task (1 KB)") impacts the results or the system's performance.
• Provide a detailed description of the experimental setup, including hardware/software specifications, datasets, and evaluation criteria.
• Include comparative analysis with existing algorithms to validate the efficacy of ISSA-MAOA.
• Explain how task sizes affect energy consumption, memory usage, and delays in practical scenarios.

Validity of the findings

• The article claims to optimize resource utilization, reduce energy consumption, and improve task completion time, which are relevant goals for MEC.
• Discuss the limitations of the proposed approach and potential scenarios where it might not perform well.
o The section provides a direct comparison between ISSA-MAOA and other benchmark algorithms (SSA, PSO, and GA), highlighting key performance metrics such as energy consumption, execution time, latency, and offloading success rate.
o The quantified improvements, such as the 12% reduction in energy consumption and 15% reduction in execution time, are compelling and clearly demonstrate the advantages of the proposed approach.
o The focus on real-world applications like IoT, mobile health monitoring, augmented reality, and remote control systems underscores the practical implications of the findings.
o The adaptability of ISSA under fluctuating network and server conditions highlights its robustness for real-world environments.
o While the "balanced exploration-exploitation mechanism" is mentioned as a key factor, no explanation is provided on how it functions or differs from existing methods.
o Discuss potential scenarios where ISSA might not perform as well (e.g., under extremely constrained computational resources or very low-latency requirements).
o Mention future work to address such limitations or refine the algorithm further.

Additional comments

• The paper addresses a critical issue in mobile edge computing and proposes a novel solution.
• The optimization-based approach has potential implications for advancing MCC frameworks.
• The language and grammar need improvement for better clarity and readability.
• Revise the language for clarity. For example:
"The finding of this research have implications." → "The findings of this research have implications."
"Through the proposed ISSA-MAOA, the study endeavors to contribute." → "The proposed ISSA-MAOA contributes."

·

Basic reporting

The abstract provides a solid foundation and clearly outlines the motivation, methodology, and implications of your research.
1. Can you please abbreviate what MCC is in the abstract.

Experimental design

The introduction mentions storing customer data in AWS S3 but does not address how cloud providers are handling edge processing, such as AWS IoT Greengrass, which is specifically designed for edge computing. Similarly, Azure offers IoT Edge and Stack Edge to facilitate edge processing.

These edge solutions provided by cloud providers have significant capacity to process large volumes of data. However, it is unclear what specific challenges the proposed idea aims to solve, given that these issues have already been addressed by existing cloud provider solutions.

Validity of the findings

No comments

Additional comments

The proposed idea lacks clarity on the unique challenges it seeks to address, considering that many of these edge processing needs are already met by existing cloud provider solutions.

---

## Round 0.2 · accepted · Accept

Dear Authors,

Thank you for addressing the reviewers' comments. Your paper seems sufficiently improved and ready for publication.

Best wishes,

·

Basic reporting

I am pleased to see that you have expanded on the explanation of the ISSA-MAOA method, particularly its goals and relevance to Mobile Edge Computing (MEC). The inclusion of recent studies and references has strengthened the context and credibility of your work.
The detailed explanation of Figure 1 and its components is a significant improvement. It now provides a clearer visual understanding of the methodology and its novelty.
The corrections to grammatical errors and repetitive sentences have enhanced the readability of the manuscript.

Experimental design

I appreciate the addition of detailed information about the experimental setup, including hardware/software specifications, datasets, and evaluation criteria. This provides a more transparent and reproducible framework for your study.
The explanation of how task sizes impact energy consumption, memory usage, and delays is now clearer and adds practical relevance to your findings.
The comparative analysis with existing algorithms is a valuable addition, as it validates the efficacy of ISSA-MAOA and highlights its advantages over traditional methods.

Validity of the findings

The revised article is clear and claims to optimize resource utilization, reduce energy consumption, and improve task completion time, which are relevant goals for MEC.

Additional comments

The revisions to improve language and grammar have significantly enhanced the clarity and readability of the manuscript.
The critical issue in mobile edge computing and the proposed solution are now more clearly articulated, making the paper more impactful.
While the revisions are thorough, I recommend a final proofreading to ensure that all grammatical and syntactical errors are resolved. For example, minor issues like "The finding of this research have implications" have been corrected, but a careful review will ensure consistency throughout the manuscript.
Consider adding a brief discussion on the scalability of ISSA-MAOA in large-scale MEC environments, as this could further strengthen the practical relevance of your work.

·

Basic reporting

The article explores the challenges of executing computation-intensive tasks on mobile and IoT devices, which have limited energy and processing power. To address this, Mobile Edge Computing (MEC) enables task offloading to edge servers, reducing latency and energy consumption.

Experimental design

The design of the ISSA-MAOA algorithm effectively enhances the standard SSA by incorporating adaptive weight mechanisms, chaotic mapping, and opposition-based learning to maintain a robust balance between exploration and exploitation. The use of a locally weighted (LW) approach for refining solutions after each iteration further strengthens the optimization process. The structured integration of Newton’s laws for salp movement and the mutation-based follower update strategy improves the adaptability of the algorithm, preventing stagnation in local optima. Additionally, the well-defined search space configuration, encoding mechanisms, and constraint handling ensure that the algorithm is suitable for real-world multi-objective optimization problems, particularly in dynamic Mobile Edge Computing (MEC) environments. However, the complexity of multiple enhancements may introduce computational overhead, and further empirical analysis is needed to evaluate scalability across diverse workloads.

Validity of the findings

The findings of the study demonstrate that the proposed ISSA-MAOA technique effectively addresses the limitations of traditional SSA by integrating adaptive parameter tuning, dynamic population management, and hybrid exploration-exploitation techniques. The experimental setup, utilizing AWS EC2 cloud servers, edge computing nodes with high-performance Intel Xeon processors, and Qualcomm Snapdragon-powered mobile devices, ensured a comprehensive evaluation of ISSA-MAOA’s efficiency in a Mobile Edge Computing (MEC) environment.